# Evaluating Therapeutic Efficacy of the Vascular Disrupting Agent OXi8007 Against Kidney Cancer in Mice [note 2]

**DOI:** 10.3390/cancers17050771

**Published:** 2025-02-24

**Authors:** Hashini I. Wanniarachchi, Regan Schuetze, Yuling Deng, Khagendra B. Hamal, Cyprian I. Pavlich, Pouguiniseli E. O. Tankoano, Caleb Tamminga, Hans Hammers, Payal Kapur, Lorena M. A. Bueno, Ricardo Rayas, Tianyuan Wang, Li Liu, Mary Lynn Trawick, Kevin G. Pinney, Ralph P. Mason

**Affiliations:** 1Department of Radiology, UT Southwestern Medical Center, 5323 Harry Hines Blvd., Dallas, TX 75390-9058, USA; hashini.wanniarachchi@utsouthwestern.edu (H.I.W.); regan.schuetze@gmail.com (R.S.); lorena.arango@utsouthwestern.edu (L.M.A.B.); ricardo.rayas@utsouthwestern.edu (R.R.); tianyuan.wang@utsouthwestern.edu (T.W.); 2Department of Chemistry and Biochemistry, Baylor University, One Bear Place #97348, Waco, TX 76798-7348, USA; yuling_deng@baylor.edu (Y.D.); khagendra_hamal@baylor.edu (K.B.H.); cyprian_pavlich1@baylor.edu (C.I.P.); elyse_tankoano1@baylor.edu (P.E.O.T.); calebtammingamed@gmail.com (C.T.); mary_lynn_trawick@baylor.edu (M.L.T.); kevin_pinney@baylor.edu (K.G.P.); 3Department of Internal Medicine, UT Southwestern Medical Center, 5323 Harry Hines Blvd., Dallas, TX 75390-8837, USA; hans.hammers@utsouthwestern.edu; 4Department of Pathology, UT Southwestern Medical Center, 5323 Harry Hines Blvd., Dallas, TX 75390-9234, USA; payal.kapur@utsouthwestern.edu

**Keywords:** vascular disrupting agent, cabozantinib, checkpoint inhibitors, kidney cancer, photoacoustic imaging, bioluminescence imaging

## Abstract

OXi8007, a water-soluble phosphate prodrug of the trimethoxyaryl-indole phenol OXi8006, was found to cause rapid acute vascular shutdown in orthotopic kidney tumors growing in mice. Photoacoustic imaging showed hypoxiation occurring within 30 min of intraperitoneal administration, and bioluminescence imaging revealed >98% vascular shutdown within 4 h. Histology confirmed the selective induction of massive hemorrhage in the tumors. The long-term twice-weekly treatment of tumor-bearing mice with OXi8007 monotherapy caused no obvious tumor growth delay, but a combination with daily cabozantinib (a frontline clinical tyrosine kinase inhibitor) caused a significant increase in the median survival time of Renca kidney tumor-bearing mice. Long-term dosing in combination with combined checkpoint inhibitors (CKIs) improved survival over CKIs alone. In vitro investigations confirmed that OXi8007 inhibited cell growth, caused microtubule disaggregation, and delayed wound closure. These results indicate the potential of vascular-disrupting agents like OXi8007 to enhance cancer treatment, prompting further investigations.

## 1. Introduction

There has been much progress in treating kidney cancer, with several new drugs being approved over the last few years. Notably, tyrosine kinase inhibitors (TKIs) such as sunitinib, pazopanib, sorafenib, and cabozantinib have extended time to progression substantially [1]. However, renal cell carcinoma remains a devastating disease, killing approximately 14,000 people annually in the United States alone. While the successive application of current therapies is prolonging survival, in many cases, ultimate relapse indicates the need for further innovation [2]. Vascular disrupting agents (VDAs) provide an attractive opportunity for selective treatment since kidney tumors are often very well vascularized.

Tumor growth beyond 1–2 mm^3^ requires a functioning vascular network. The neovasculature of tumors is often undergoing rapid proliferation, while it is poorly differentiated, lacking pericyte support, and exhibiting increased permeability. It is also generally more responsive to angiogenic cell signaling [3,4]. In contrast, the endothelium of normal blood vessels is largely quiescent. Consequently, tumor-associated vasculature offers a unique, potentially selective target for anticancer therapy. While the concept of vascular targeting was proposed around 30 years ago [5,6,7,8], there is a resurgence of interest [9,10,11,12,13], driven in part by superior next-generation VDAs and potential opportunities for synergy with new treatments such as immunotherapy, whereby cold tumors may be inflamed through enhanced antigen presentation [14]. The goal of VDA treatment is to cause the rapid and widespread disruption of existing tumor-associated vasculature, leading to selective blood flow shutdown in the tumor, the induction of hypoxia, and amplified tumor necrosis [8,15]. Most VDAs inhibit microtubule formation by binding to the colchicine site on the tubulin heterodimer in endothelial cells lining tumor-associated vasculature [16]. The mechanism of action of OXi8007 in activated endothelial cells was presented in detail previously [16]. Briefly, the phosphate prodrug OXi8007 is cleaved by a non-specific phosphatase to OXi8006, which enters endothelial cells and binds to tubulin, resulting in net microtubule depolymerization and RhoA activation. A series of kinases activate non-muscle myosin II, which results in actin bundling and stress fiber formation. Endothelial cells round up and detach from the microvascular lining. Microtubule disaggregation leads to rapid morphological changes, resulting in massively increased vessel permeability, endothelial cell detachment, ischemia, and hypoxia. Treatment with VDAs is characterized by cell death in the central tumor, even in large tumor masses, which otherwise often resist treatment.

Several pre-clinical studies have demonstrated the activity of previous VDAs (e.g., CA1P (also called combretastatin A1 diphosphate or OXi4503)*,* BNC105P, DMXAA (also called dimethylxanthenone acetic acid, Vadimezan, or ASA404), and ZD6126 (also called ANG 453 or *N*-acetylcochinol-*O*-phosphate)) against kidney cancer [17,18,19,20], but sustained therapeutic efficacy in terms of tumor growth delay required a combination with additional treatments (drugs or radiation). Effective combination was reported with the anti-angiogenesis agent bevacizumab [17], the mTOR inhibitor everolimus [18,20], or tyrosine kinase inhibitors (TKIs) such as sunitinib [21] or pazopanib [18], yielding tumor growth delay in Renca-luc and Caki-1 tumors. Nonetheless, there is a clear need for a more effective VDA, and here we report investigations with OXi8007 (Figure 1), a water-soluble phosphate prodrug of OXi8006, an indole-based molecule structurally inspired by combretastatin A-4 (CA4) [22]. Pinney et al. synthesized the trimethoxyaryl-indole phenol OXi8006, which acts as an inhibitor of tubulin assembly in vitro (IC_50_ = 1.1 μM), and it was found to be strongly cytotoxic against human cancer cell lines (e.g., GI_50_ = 32 nM against MDA-MB-231 cells, and GI_50_ = 41 nM against activated human umbilical vein endothelial cells (HUVECs)) [22,23,24]. The water-soluble prodrug OXi8007 causes effective vascular disruption in several tumor types growing in mice, including human PC3 prostate [22] and MDA-MB-231 breast tumors [15,25,26], as well as mouse Renca kidney tumors [15]. The prodrug OXi8007 undergoes rapid in vivo conversion to the active compound OXi8006 through the action of non-specific phosphatase enzymes. ^19^F MRI revealed progressive hypoxiation in MDA-MB-231 tumors over a period of 2 h following the IP administration of OXi8007 at a dose of 350 mg/kg [26].

A common side effect observed in patients and pre-clinical animal studies of combretastatin was hypertension, which could be mitigated with calcium channel blockers or hydralazine [27,28,29]. As part of the current evaluation of OXi8007, we examined potential cardiovascular effects.

Many modalities are becoming available for small animal imaging [30], and vascular shutdown (ischemia) has been evaluated using various techniques. Early studies examined histological slices of tumors following the administration of a perfusion indicator, such as Hoechst 33442, indocyanine green (ICG), colored microbeads, or radiolabeled markers [15]. More recently, noninvasive imaging has revealed VDA activity based on radionuclide approaches, MRI, ultrasound, fluorescence, and laser Doppler [26,31,32,33,34,35,36]. The availability of luciferase-transfected cells has provided a particularly effective means of evaluating tumor perfusion, since ischemia inhibits the delivery of luciferin substrate to the tumor cells, leading to significantly reduced bioluminescent light emission [15,37]. Quite recently, photoacoustic methods have become available [15,38]. Multispectral optoacoustic tomography (MSOT) directly reveals relative spatial and temporal variations in oxy- and deoxyhemoglobin concentrations without the need for exogenous reporter molecules or cell transfections [38,39,40]. MSOT is particularly suitable for evaluating the sequelae of VDA activity, with recent reports examining brain, prostate, head and neck, breast, lung, and kidney tumors [15,41,42,43,44].

Anticipating the need for combination treatment, we examined the potential opportunity to enhance the response to current treatments for kidney cancer. Specifically, the TKI cabozantinib targets the c-MET receptor, AXL, and VEGFR-2, and it is the first-line treatment option for metastatic renal cell carcinoma (RCC) [45]. Since VDA-induced hypoxia should further upregulate VEGFR-2, an additional benefit was anticipated. Immunotherapy based on checkpoint inhibitors (CKIs) has shown remarkable success in treating various cancers, including RCC, where many patients show long-term control with multiple previous lines of failed chemotherapy [46,47,48]. Recently, the combination of nivolumab (PD-1 blockade) plus ipilimumab (CTLA-4 blockade) was approved for intermediate and poor-risk patients with advanced RCC based on the CheckMate 214 study in the first-line setting [47,49]. However, not all patients respond, and it is recognized that additional therapy will be needed for broader success.

An established model for investigations of immunotherapy in RCC is Renca in BALB/c mice [50,51,52] with past studies of the mechanistically distinct VDA DMXAA [20]. Our current investigation of OXi8007 is summarized in Graphical abstract. Here, we have characterized the effects of OXi8007 on Renca cells in terms of cytotoxicity, cytoskeletal structure, cell morphology, cell growth, and cell mobility in culture. Acute vascular disruption was examined in orthotopic Renca-luc tumors growing in BALB/c mice using BLI and MSOT. Tumor control was evaluated in orthotopic Renca-luc tumors in combination with cabozantinib or checkpoint inhibitors (anti-PD-1 and anti-CTLA-4). To demonstrate further generality, acute vascular disruption and therapeutic control were examined in orthotopic XP258 human tumor xenografts, which are noted to be resistant to treatment with sunitinib, the HIF-2α antagonist PT2399, and the ubiquitin-activating enzyme TAK-243 [53,54].

## 2. Materials and Methods

### 2.1. Cell Culture

Renca mouse kidney cancer cells (ATCC) and Renca-luc cells [55] were grown in RPMI-1640 media supplemented with 10% qualified FBS, sodium pyruvate, MEM non-essential amino acids, *L*-glutamine, and Pen/Strep solution. Cells were authenticated using the FTA Sample Collection Kit (ATCC, 137-XV) through the Mouse Cell Authentication Service at ATCC.

#### 2.1.1. Cell Toxicity: Renca Sulforhodamine B (SRB) Assay

IC_50_ values for OXi8006, OXi8007, and combretastatin A-4 (CA4; positive control) were determined for Renca cells using the SRB assay [56,57,58]. In two independent trials of Renca and Renca-luc cells were plated in triplicate at 8000 cells/well into 96-well plates and incubated for 24 h at 37 °C in a humidified incubator with 5% CO_2_ followed by a 48 h treatment period at ten-fold dilutions of the drug.

#### 2.1.2. Microtubule Disruption in Renca Cells Treated with OXi8007

Renca cells were plated in cell culture-treated 96-well plates (Grenier Bio-One North America Inc., Monroe, NC, USA) at 2000 cells/well and allowed to adhere for 24 h before treatment. Hoechst 33342 (1 μg/mL) in complete RPMI 1640 medium was used to stain the cell nuclei for 15 min at 37 °C. The medium was gently removed, and the cells were stained with fresh medium containing 1X ViaFluor 488 (Biotium Inc., Freemont, CA, USA) live cell microtubule stain with 25 μM of verapamil hydrochloride (Biotium), an efflux pump inhibitor to improve probe retention, for 30 min at 37 °C in the dark. The medium was removed to reduce background fluorescence, followed by two wash cycles with fresh medium. Fresh medium containing 25 μM of verapamil hydrochloride was added to the stained cells prior to imaging. Images of the microtubule cytoskeleton and nuclei were collected over 3 h using a Biotek Lionheart microscope ELx800 in fluorescence mode with a 20× objective.

#### 2.1.3. Wound (Scratch) Assay of OXi8006 and OXi8007

Renca cells were plated at a seeding density of approximately 3 × 10^5^ cells per well in a 6-well plate and allowed to reach confluency before treatment. A scratch was made with a sterile 10 µL pipette tip in each well before removing the medium and any detached cells. The wells were treated with varying concentrations of OXi8006 or OXi8007 (0, 0.5, 1.5, 2.5, 5, and 10 µM), and the wound closure was monitored with an automated live cell imaging microscope (BioTek Lionheart ELx800, BioTek Instruments Inc., Winooski, VT, USA) for 48 h. Hoechst 33342 was added to each well at a final concentration of 1 µg/mL and incubated for 15 min at 37 °C to stain the cell nuclei. Gen5 version 3.14 software was used for cellular and statistical analysis of the wound closure data.

### 2.2. Mouse Models and Imaging Methods

The animal procedures were approved by the UT Southwestern Medical Center Institutional Animal Care and Use Committee under APNs 2017-102152 and 2018-102344-CORE. Renca-luc orthotopic tumors were induced by inserting tumor tissue (~1 mm^3^) harvested from a donor mouse into the right kidney of syngeneic BALB/c mice (age 8 weeks, male and female; Envigo, Indianapolis, IN, USA), as described previously [44]. Orthotopic XP258 tumors (derived from a 51-year-old patient with pT3aN0M1 clear cell RCC) were obtained from the UT Southwestern Kidney Cancer SPORE [53,54] and maintained by serial passage. Tumors were induced by inserting tumor tissues from a donor tumor into the left kidney capsule of female NOD/SCID mice (Envigo) [59].

#### 2.2.1. Pharmacokinetic Studies of OXi8007 in Renca Tumor-Bearing BALB/c Mice

Plasma and tissues were harvested at various time points (20 min to 16 h) from Renca-luc tumor-bearing BALB/c male mice following the administration of OXi8007 IP. Mice treated with 250 mg/kg were sacrificed at 20 min (*n* = 3), 1 h (*n* = 4), 2 h (*n* = 4), 4 h (*n* = 4), 8 h (*n* = 4), and 16 h (*n* = 4). Mice were anesthetized, exsanguinated, the tissues were blotted, and then they were weighed and frozen in liquid nitrogen. Homogenized plasma and tissue samples were extracted with acetonitrile with a bead mill; internal standard (CA4 or tri-labeled ^13^C-OXi8006) was added to each sample to reach a final concentration of 10 ppb. Samples were centrifuged twice to remove precipitated proteins (16,260× *g* for 10 min at 4 °C). Clear supernatant was transferred into MS vials, and formic acid was added to a final concentration of 0.1% for LC-MS analysis. OXi8007, OXi8006, and metabolites were determined using a Thermo Orbitrap Exploris 120 or Thermo Q-Exactive Orbitrap and LC with a BDS Hypersil C18 column (100 mm × 2.1 mm, particle size 3 µm). Mobile phase A consisted of 0.1% *v*/*v* formic acid in water, and mobile phase B was acetonitrile, which was mixed by the binary pump with an elution gradient at a flow rate of 0.6 mL/min. Data were analyzed using Xcalibur 4.6 software based on exact masses, and Phoenix WinNonlin 8.5.0 software was used to determine the pharmacokinetic parameters based on a non-compartmental model. Standard curves for OXi8007, OXi8006, and internal standards were determined frequently.

#### 2.2.2. Blood Pressure Assessment

Blood pressure was measured using a CODA system (Kent Scientific, Torrington, CT, USA) on 5 adult male BALB/c mice. Mice were anesthetized with 2% isoflurane in oxygen at 1 dm^3^/min and placed on the CODA IR heating pad with a feedback rectal temperature probe and nosecone to maintain anesthesia. An occlusion cuff and VPR cuff were placed on the tail. The baseline measurement was recorded over 30 min, then OXi8007 (250 mg/kg or 325 mg/kg) was administered IP and the blood pressure measurement resumed for 45 min. As positive and negative controls, hydralazine (1 mg/kg) or saline, respectively, were injected IP.

#### 2.2.3. Bioluminescence Imaging (BLI)

Anesthesia was induced with 1–3% isoflurane, and the mice were placed on the warmed imaging stage of an IVIS Spectrum^®^ small animal imaging system (Xenogen/PerkinElmer Inc., Waltham, MA, USA) with continuous exposure to 1 to 2% isoflurane, as described previously [60]. On successive occasions, mice were administered 75 µL of *D*-luciferin (sodium salt; Gold Biotechnology, St. Louis, MO, USA) SC in the fore back neck region as a 40 mg/mL solution in 0.9% saline. The BLI signal was quantified as the total flux using Living Image Software 4.3.7. Fresh luciferin was administered 4 and 24 h after drug administration to reveal acute changes in tumor perfusion.

#### 2.2.4. Multispectral Optoacoustic Tomography (MSOT)

For the multispectral optoacoustic tomography (MSOT), an iThera InVision 256-TF device with seven wavelengths (700, 730, 760, 780, 800, 820, and 875 nm) was used [44]. Mice were shaved and depilated around the tumor region. The anesthetized mouse (1–2% isoflurane) was placed in the imaging chamber (34 °C) for 10 min to reach thermal equilibrium before imaging a single slice showing the largest cross-section of the tumor. An oxygen gas breathing challenge was performed to assess the dynamic changes in blood oxygenation. Images were generated using ViewMSOT version 4.2 with a back projection-based reconstruction algorithm followed by built-in fluence correction. Oxy- and deoxyhemoglobin were resolved using a linear regression approach.

Oxygen saturation (sO_2_) maps were analyzed using MATLAB 2022b (MathWorks Inc., Natick, MA, USA), and the images were visualized with Fiji [61]. The percentage of hemoglobin saturation (sO_2_) was calculated as follows:sO_2_ = [HbO_2_]/[Hb] + [HbO_2_](1)

Blank pixels represent areas lacking a detectable hemoglobin signal. The average sO_2_^MSOT^ was calculated pixelwise in each ROI for the air and oxygen breathing periods as sO_2_^MSOT^ (Air) and sO_2_^MSOT^ (O_2_), and the response to the oxygen gas challenge was as follows:∆sO_2_ = sO_2_ (O_2_) − sO_2_ (Air)(2)

Hemoglobin oxygen saturation data (sO_2_) were compared using a paired *t*-test and a two-sample *t*-test (MathWorks Inc.).

### 2.3. Treatment Response to OXi8007

OXi8007, synthesized by the Pinney Laboratory using published procedures [22], was dissolved in saline, verified to be in the pH range 7.3 to 7.5, and dosed at 250 mg/kg (IP). Cabozantinib (in PEG400 and ultrapure H_2_O (Milli-Q)) was administered at 3 mg/kg by oral gavage. Anti-PD-1 (10 mg/kg, Bio X Cell, Lebanon, NH, USA) and anti-CTLA-4 (5 mg/kg, Bio X Cell, Lebanon, NH) solutions were administered (IP) on the day before OXi8007 twice during the first week.

#### 2.3.1. Acute Response to OXi8007

Acute effects of OXi8007 were examined on Renca-luc tumor-bearing mice using BLI (n = 8) and MSOT (n = 6) by comparing images at the baseline [pre (0 h)] and following OXi8007 administration (IP) at 4 h and 24 h. Dynamic BLI images were acquired every minute over the course of 15–20 min immediately following luciferin injection. For MSOT, the same mice underwent a gas breathing challenge for 5 min in air (21% O_2_), 7 min in O_2_ (100% O_2_), followed by 3 min in air at 0 h, 4 h, and 24 h post-treatment.

#### 2.3.2. Drug Combination Therapy

We evaluated the longitudinal effects of OXi8007 alone and in combination with drugs currently favored for treating kidney cancer. For Renca-luc tumor-bearing mice, BLI was performed at least once a week to estimate tumor size, and mice were weighed weekly to adjust the dose as needed. Two treatment cohorts examined syngeneic orthotopic Renca-luc tumors in male BALB/c mice. Treatment started when the tumors reached about 1 × 10^6^ photons/s peak total flux on BLI. The first cohort examined OXi8007 with cabozantinib. Mice were divided into four groups. Mice were assigned to groups as soon as there were sufficient to add the same numbers of animals to each group. The specific assignment was defined using the random number function in Excel. As such, each group included both faster- and slower-growing tumors. Group 1, the control (n = 7), was treated with 100 mL saline IP daily; Group 2 (OXi8007) (250 mg/kg, IP, n = 9) was treated twice weekly (Monday and Friday); Group 3 (cabozantinib) (n = 9) was treated by oral gavage daily (7 days a week) with 3 mg/kg; and Group 4 (combination cabozantinib + OXi8007; n = 9) has the same dosing regimens as those for the monotherapy groups, and OXi8007 was administered at least 2 h after cabozantinib on days when both were administered. The second cohort was treated with CKIs and OXi8007 using four groups: Group 5 (OXi8007 + anti-PD-1 (10 mg/kg) + anti-CTLA-4 (5 mg/kg); n = 4), Group 6 (saline + anti-PD-1+ anti-CTLA-4; n = 3), Group 7 (saline + anti-PD-1; n = 3), and Group 8 (saline + anti-CTLA-4; n = 4). CKI antibodies were administered thrice weekly (M, W, F) for the first two weeks after the first treatment with OXi8007, which was administered twice weekly. A third cohort matched Cohort 1, but with orthotopic XP258 human tumor xenografts, where treatment started 9–14 days post-tumor implantation, based on historic growth rate expectations. Initial tumor growth was confirmed by palpation, and the mice were assigned to groups such that the first mouse with a confirmed tumor was assigned to Group 5, next to Group 6, etc., ensuring that all groups had both faster- and slower-growing tumors. Mice were weighed weekly and sacrificed if they lost more than 20% of their body weight, showed severe toxicity, or if a tumor reached 10% of the mouse’s body weight. Tumor size (BLI signal) and mouse body weights were compared using an ANOVA and multiple comparisons at a significance level of *p* < 0.05. Kaplan–Meier survival curves were compared using a non-parametric log-rank test using SAS software (SAS Institute Inc., SAS Viya for learners 4, Cary, NC, USA).

## 3. Results

### 3.1. OXi8007 Mechanism and Cell Studies

Renca and Renca-luc cells were found to be similarly sensitive to OXi8006 and OXi8007 but showed greater sensitivity to CA4 (Table 1). Treatment of Renca cells with OXi8007 (2.5 and 5 µM) resulted in the rapid progressive loss of microtubule structure with a consequent reduced proliferation (Figure 2 and Appendix A). OXi8006 had little effect up to 2 µM, but at 4 µM, the cells appeared stressed and lost their normal cell shape at the early time points (<24 h), though the cells recovered and started to grow again.

Renca cells showed a low propensity for migration, with only 30 to 40% wound closure over 48 h. When exposed to concentrations of OXi8006 or OXi8007, ≥ 2.5 µM wound closure was significantly depressed, particularly for OXi8007 at later times (Figure 3 and Appendix A). This is consistent with the potent effect of OXi8007 on Renca cell microtubule disruption (Figure 2), since microtubules control directional cell migration. At higher OXi8007 concentrations, the inhibition of cell proliferation also contributed to the inhibition of wound closure.

#### 3.1.1. Pharmacokinetic Studies of OXi8007 in Renca-Luc Tumors Bearing BALB/c Mice

In the plasma and tumor, prodrug OXi8007 was present at higher concentrations than OXi8006, while in other tissues, OXi8006 was higher (Figure 4 and Table 2 and Table 3). Plasma had the highest concentrations of both OXi8007 and OXi8006 at the first measured time point with monotonic decreases over the next 16 h (T_1/2_ OXi8007 = 49 min, OXi8006 = 119 min). Tissue concentrations were about 100-fold lower than plasma, and most tissues were much lower than tumor. Several metabolites were detected, including OXi8006-glucuronide in all tissues (Figure 1 and Figure 4, and Table 4).

#### 3.1.2. Blood Pressure Fluctuation upon Administrating OXi8007

At 250 mg/kg, OXi8007 had a minimal effect on mouse blood pressure (systolic, diastolic, or mean; Figure 5). Baseline blood pressure was stable over a period of 25 min. A transient spike was seen immediately following OXi8007 administration, but it rapidly returned to the baseline. A larger transient change was observed following 325 mg/kg IP (Appendix A). Administering saline caused no effect, but as expected, hydralazine caused a rapid drop in blood pressure of about 30% or Δ = 40 mmHg (systolic), Δ = 31 mmHg (diastolic), and Δ = 34 mmHg (mean), respectively (Appendix A).

### 3.2. Acute Renca-Luc Tumor Response to OXi8007

The BLI signal from orthotopic Renca-luc tumors (Figure 6A) reached a maximum within 6 min, followed by progressive diminution (Figure 6B). A significantly reduced BLI signal (<2% baseline, *p* < 0.001) was observed following the administration of fresh luciferin to the group of 8 mice 4 h after OXi8007 (250 mg/kg, IP), with slight recovery at 24 h (Figure 6C). Histology confirmed extensive intratumoral hemorrhage within 4 h post-OXi8007 treatment (Figure 6D).

Photoacoustic tomography showed heterogeneous baseline vascular oxygenation in the Renca-luc tumors (Figure 6F), which was stable and responded with a rapid significant increase in sO_2_^MSOT^ in response to an oxygen gas breathing challenge (*p* = 0.0012; Figure 6E,F). Following the injection of OXi8007 (350 mg/kg, IP in situ using a catheter), vascular hypoxiation was observed over a period of about 30 min settling on a plateau significantly below the oxygenation observed with breathing air at baseline (Figure 6F). Meanwhile, the contralateral control kidney and spine muscle showed significantly increased vascular oxygenation with the O_2_ gas breathing challenge but no obvious response to OXi8007 (Figure 6G). The acute effects of OXi8007 on Renca-luc tumors were confirmed across a group of mice (n = 6; Figure 6H). Tumor oxygenation often showed a bimodal distribution and a modest, but significant, response to the oxygen gas breathing challenge (Figure 6G,H). Four hours after OXi8007 (250 mg/kg), the histogram of baseline oxygenation showed a distinct left shift, and then the responses to O_2_-breathing were much smaller. In some tumors, there was further hypoxiation in response to oxygen breathing (Figure 6H). This lack of increase continued at 24 h. The contralateral kidney and spine muscle each showed significant enhancement in vascular oxygenation at baseline, and the response was similar at 4 and 24 h.

### 3.3. Combination Therapy

At the time of treatment, all tumor groups indicated similar BLI signals (about 10^6^ photons/s). Bioluminescence indicated that control Renca-luc tumors grew with a volume doubling time of 15 days (Figure 7A). On day 21, both Groups 3 (cabozantinib) and 4 (cabozantinib + OXi8007) were significantly smaller than the control tumors (*p* < 0.01), and on day 35, all of the treated groups had significantly smaller tumors (less light emission) than the control group (*p* < 0.005). Ultimately, mice treated with OXi8007 alone showed no overall survival benefit (*p* = 0.053, non-parametric Mantel–Cox log-rank test, Figure 7B). Treatment with cabozantinib alone showed a significant tumor growth delay (*p* < 0.0001 vs. control) and an extended median survival time of 64 days (*p* < 0.0001 vs. control). The median survival of Group 4 was significantly extended to 78 days (*p* < 0.0001 vs. Group 3). Groups 3 and 4 showed a survival benefit compared to the control group (*p* = 0.04 and *p* = 0.0069) or OXi8007 mono therapy group (*p* < 0.0001, and *p* = 0.0013). Histology indicated that tumors from each group showed extensive necrosis at sacrifice (Figure 7C). The extent of viable tumor was comparable between all three of the treated groups and less than that observed in the control mice (Figure 7C–E). Mouse body weight was similar for each group at baseline. Over the course of the treatment, cabozantinib alone (3 mg/kg daily) caused significant weight loss compared with the untreated control mice (*p* < 0.005) (Appendix A). Mice treated with OXi8007 alone showed a significant loss in body weight at week 7 compared with baseline (*p* = 0.0007).

Bioluminescence imaging indicated that combined treatment with OXi8007 and paired CKIs (Group 5 anti-PD-1 + anti-CTLA-4) yielded no obvious tumor growth delay, but there was a significantly enhanced tumor survival (median 58 days) compared with either antibody alone or the combined antibodies (Figure 8A,B) (Group 5 vs. Group 7, *p* = 0.025) (Group 5 vs. Group 8, *p* = 0.018). Treatment with anti-CTLA caused significant body weight loss compared with each of the other treatment groups by week three (Appendix A). Upon sacrifice and autopsy, metastases were often seen in the lung and liver in all of the treatment groups of Renca-luc tumors, but rigorous quantification was not possible.

### 3.4. Effects of OXi8007 on Patient-Derived XP258 Tumors

Distinct heterogeneity in vascular oxygenation was observed in the XP258 tumors with the periphery being much better oxygenated (Figure 9). The tumor is readily identified in the grey scale single wavelength image (Figure 9A) and it responded to an oxygen gas breathing challenge (Figure 9B,E). Four hours post-treatment with OXi8007, the tumor was less well oxygenated and less responsive to the oxygen gas breathing challenge. For a group of six tumors, the mean vascular oxygenation changed little, but the oxygen gas breathing challenge caused hypoxiation (Figure 9C,D), and extensive hemorrhage was observed microscopically (Figure 9F).

The treatment of XP258 tumors with cabozantinib or OXi8007 alone gave significantly enhanced survival compared with the controls (Figure 10A; *p* = 0.0025 and *p* = 0.018), though combined treatment offered no additional benefit. However, tumor weights at the time of sacrifice following long-term therapy indicated that mice receiving OXi8007 (monotherapy or combined) had much smaller tumors (Figure 10B). Tumors receiving OXi8007 as monotherapy or in combination showed extensive necrosis and focal hemorrhage (Figure 10C–E. No metastases were observed in mice bearing the XP258 tumor. No significant body weight changes were observed for any of the groups of NOD/SCID mice bearing XP 258 tumors treated with OXi8007 and/or cabozantinib (Appendix A).

## 4. Discussion

OXi8007 caused acute vascular shutdown within 4 h in both syngeneic Renca-luc and human XP258 orthotopic kidney tumors in mice following a dose of 250 mg/kg IP. As a monotherapy, there was no obvious tumor growth delay or survival benefit in response to a twice-weekly dosage over several weeks in Renca-luc tumors, but XP258 human tumor xenografts exhibited a significant survival benefit. Twice-weekly OXi8007 increased the median survival of RENCA tumor-bearing mice when added to daily cabozantinib, or treatment with CKIs. OXi8007 was well tolerated and appears worthy of further exploration.

BLI revealed acute vascular shutdown within 4 h of administering OXi8007 at 250 mg/kg IP (Figure 6), closely matching many other VDAs such as CA4P, BPR0L075, and KGP265 [15,37,44,60,62]. MSOT corroborated this observation in the Renca–luc tumors and importantly allowed for observations in the untransfected XP258 human tumor xenografts. MSOT revealed acute hypoxiation in real time over a period of 30 min after the administration of OXi8007 via an indwelling peritoneal catheter (Figure 6E). The pharmacokinetic assessment showed that OXi8007 reached a maximum concentration in Renca-luc tumors within 1 h following IP administration (Figure 4), while plasma had a concentration of about 450 µg/mL, which decreased to 10 µg/mL within 4 h. Not surprisingly, concentrations in tissues were about 100-fold lower, but 15 ng/mg in the tumor would be consistent with 32 µM, thereby exceeding the IC_50_ observed for cells. Indeed, the concentration appears to exceed the IC_50_ up to 4 h. In most tissues, OXi8006 was found at a higher concentration than the prodrug OXi8007, potentially because it must be dephosphorylated to enter cells. Conversely, in tumors, OXi8007 was elevated, which may reflect leakage from blood vessels into hemorrhagic tissue. This is the first reported pharmacokinetic analysis regarding the LC-MS of OXi8007, and the methodology was closely analogous to previous work exploring CA4P [63]. Several additional metabolites were observed, including glucuronides, desmethoxy molecules, and N-acetylated species, which were also reported for CA4P [63,64].

A challenge with CA4P in the clinical trials was hypertension, although this was controlled using low doses of hydralazine [64,65]. Here, we examined potential hypertension with respect to OXi8007. A minor temporary elevation in blood pressure occurred, which lasted less than 20 min, but it was somewhat greater at a dose of 325 mg/kg (Appendix A). Reassuringly, the injection of saline indicated very stable blood pressure, while administering hydralazine at 1 mg/kg caused rapid hypotension (Δ BP~35 mmHg), in line with previous reports [66,67].

The orthotopic Renca-luc tumors grew with a mean volume doubling time of about 15 days. Treatment twice weekly with OXi8007 (250 mg/kg, IP) caused a tumor growth delay over the first 40 days, but by day 50, there was no benefit in terms of tumor volume (based on emitted light) or overall survival (Figure 7). Cabozantinib alone (administered daily at 3 mg/kg) gave a significant tumor growth delay, and combination with OXi8007 extended median survival time. At autopsy, all tumors showed extensive necrosis and very limited viable tissue (Figure 7C). In XP258 tumors, OXi8007 alone provided a significant tumor growth delay, matching cabozantinib alone, but it provided no additional benefit when combined (Figure 10). We are aware of only two previous investigations of OXi8007 as a long-term therapeutic in mice. Dalal and Burchill found no tumor growth delay in Ewing sarcoma in mice dosed at 200 mg/kg twice weekly [68]. Mice bearing subcutaneous MDA-MB-231 tumors showed a significant tumor growth delay, as reported in a patent application [69]. Others have examined the response of subcutaneous Renca tumors to combined cabozantinib and anti-Ly6G or anti-CD8 antibody or the inhibition of lung metastasis following the resection of primary orthotopic tumors in mice [70,71]. They each used 10 mg/kg, as opposed to 3 mg/kg daily used here. Subcutaneous tumors were 50% smaller on day 14, when mice were terminated [70], as opposed to the much longer-term study we conducted that generated a median survival of 64 days. It should be noted that cabozantinib is often given as a dose of 30 mg/kg by oral gavage daily in mice [72]. However, cabozantinib is associated with considerable side effects in many patients (notably diarrhea), and we chose to administer a lower dose (3 mg/kg) here, as also recently reported by others [73]. Nonetheless, a significant loss in body weight was observed in the mice over 7 weeks compared with the untreated controls (Appendix A). We also found a loss in body weight for mice receiving anti-CTLA4 compared with other cohorts (Appendix A). OXi8007 alone showed no significant difference in body weight compared with other cohorts, but the mice themselves did significantly lose weight compared with the pre-treatment baseline (~11%, *p* < 0.007; Appendix A).

Several VDAs have been evaluated in clinical trials, though to date, none have received FDA approval [15]. CA4P (fosbretabulin or Zybrestat), DMXAA (Vadimezan), and ZD6126 underwent Phase 2 clinical trials, and CA4P and AVE8062 (Ombrabulin) reached Phase 3 [74,75], but they failed to reach the requisite therapeutic targets. Several were also noted to cause cardiovascular effects, such as hypertension. CA4P was granted orphan drug status by the European Medicines Agency (EMA) and Federal Drug Administration (FDA) [16]. Only one clinical trial using VDAs appears to be active: currently “Modulation Of The Tumour Microenvironment Using Either Vascular Disrupting Agents or STAT3 Inhibition in Order to Synergise With PD1 Inhibition in Microsatellite Stable, Refractory Colorectal Cancer (MODULATE)”, which examines BNC105P and is organized by the Australasian Gastro-Intestinal Trials Group [76]. Here, we found that anti-PD1 alone or combined with anti-CTLA-4 delayed tumor growth significantly compared with anti-CTLA-4 alone (Figure 8), and adding OXi8007 to the combined regimen provided significantly enhanced overall survival, though no cures. The combination severely reduced the extent of viable tumor observed at sacrifice (Figure 8C–E).

OXi8007 was originally designed, synthesized, and developed by us (Pinney lab), and we note that it is now listed as being commercially available (MedChem Express). Likewise, we note that many propriety VDAs are similarly available for research, though not to be used in patients.

Dose and dosing frequency can significantly influence efficacy. It has been reported that CA4P was ineffective in terms of tumor growth delay at a standard dose of 100 mg/kg twice weekly in mice, but when applied at 50 mg/kg BID, a tumor growth delay was observed [77]. It will be interesting to explore other dosing schedules for OXi8007 in the future. Efficacy might also be enhanced by delivery and slow release from nanoparticles [78]. Other prodrugs may provide advantages such as enzyme- or hypoxia-activated molecules [79,80]. It will be important to examine tumors in other species, noting that mouse vasculature appears to be particularly resistant to vascular disruption, e.g., CA4P is typically applied at 120 mg/kg when treating tumor-bearing mice [3], while 30 mg/kg was found to be effective in tumor-bearing rats [43]. There are also opportunities to modify the molecular structure; the analogous benzosuberene (KGP265) yields similar vascular disruption at about 10 mg/kg [44], but the ultimate therapeutic window between minimum effective dose and maximum tolerated dose limits applications. In that respect, CA4P remains attractive since it is typically effective at 100 mg/kg in mice, which is about one-fifth of the maximum tolerated dose [81].

## 5. Conclusions

In summary, the results confirm the rapid acute effects of OXi8007 on tumors in terms of vascular disruption and hypoxiation. In some cases, OXi8007 yielded a tumor growth delay as a monotherapy, and in other cases, it enhanced combination treatment. It is reassuring that OXi8007 caused no overt toxicity in adult mice when dosed at 250 mg/kg IP twice weekly up to 7 weeks. We are currently expanding investigations into other tumor types and drug combinations and schedules to establish optimal efficacy.

## 6. Patents

National Center for Biotechnology Information (2025). PubChem Patent Summary for WO-0168654-A2, Tubulin binding ligands and corresponding prodrug constructs. Retrieved 17 January 2025, from https://pubchem.ncbi.nlm.nih.gov/patent/WO-0168654-A2, Kevin G. Pinney, Feng Wang, and Mallinath Hadimani, Indole-Containing and Combretastatin-Related Anti-Mitotic and Anti-Tubulin Polymerization Agents, United States Patent (US 6,849,656 B1), Issued 1 February 2005 and Kevin G. Pinney, W.; Wang; F.; Hadimani; M.; Mejia; M.D.P. Indole-Containing Compounds with Anti-tubulin Vascular Targeting Activity. US 2007/0082872 A1, 2007.

## Figures and Tables

**Figure 1 cancers-17-00771-f001:**
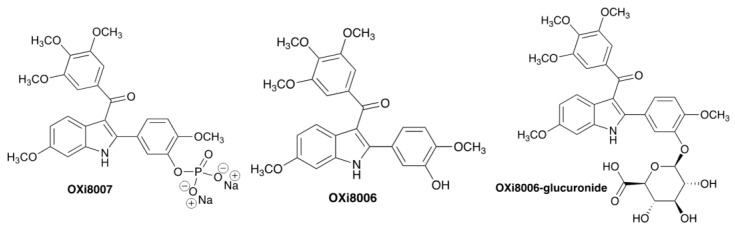
Molecular structures. Hydrolysis of OXi8007 by abundant non-specific phosphatases yields the active agent OXi8006, which was subject to glucuronidation.

**Figure 2 cancers-17-00771-f002:**
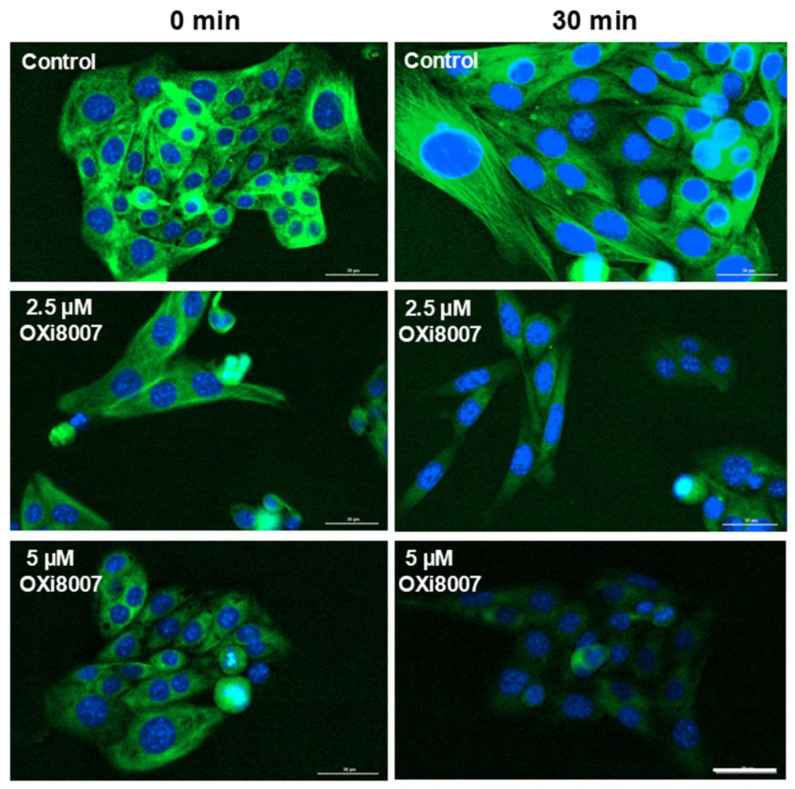
Microtubule disruption in Renca cells following treatment with OXi8007. Images show plates of Renca cells after 30 min exposure to OXi8007 (0, 2.5, or 5 µM). Stained with ViaFluor 488 live cell microtubule stain (green) and Hoechst 33342 blue (nuclei). Scale bar is 30 µm. Additional images in Appendix A.

**Figure 3 cancers-17-00771-f003:**
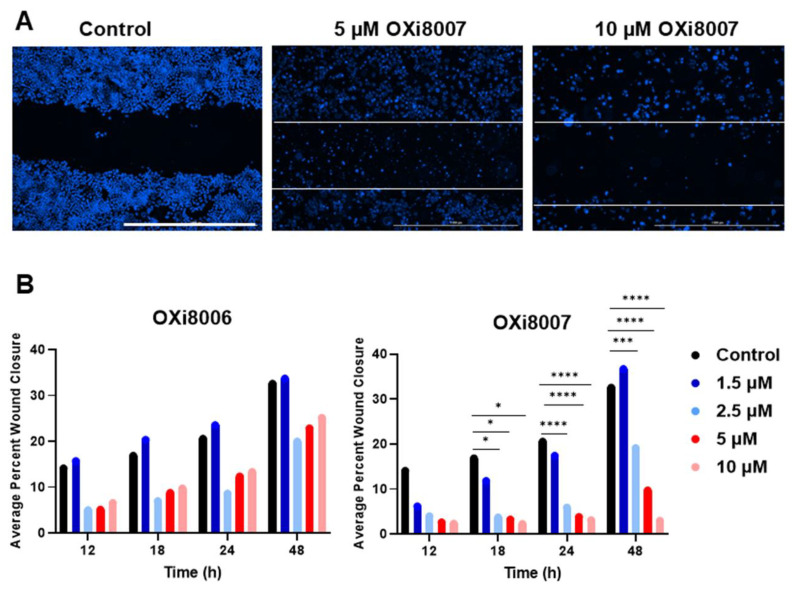
Wound healing assay. (**A**) Renca cells treated with OXi8007 (0, 5, or 10 µM) for 48 h. Blue: Hoechst 33342 stain for the nuclei. Scale bar 1000 µm. (**B**) Average percent wound closure of Renca cells treated with varying concentrations of OXi8006 or prodrug OXi8007. Significance was calculated using a one-way ANOVA with *p* < 0.05 *; *p* < 0.001 ***; and *p* < 0.0001 ****. Additional images in Appendix A.

**Figure 4 cancers-17-00771-f004:**
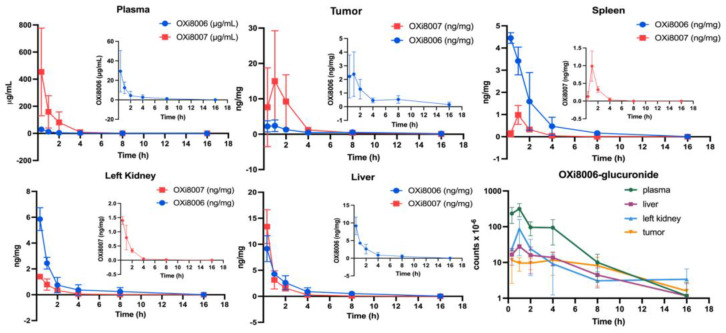
Uptake and clearance of prodrug OXi8007, as well as OXi8006 and product of glucuronidation following administration of OXi8007 (250 mg/kg, IP).

**Figure 5 cancers-17-00771-f005:**
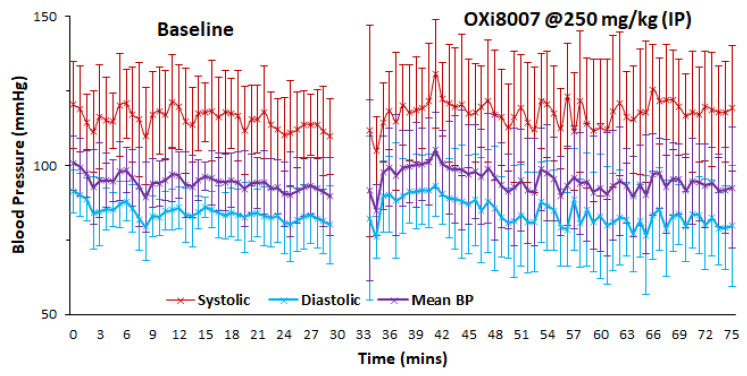
Systolic (red), diastolic (blue), and mean (purple) blood pressure for a group of 5 male BALB/c mice observed continuously before and after administration of 250 mg/kg OXi8007 IP. Baseline was stable (systolic 115 ± 3 mmHg, diastolic 84 ± 2 mmHg, and mean 94 ± 2 mmHg.) Following the administration of OXi8007, the blood pressure remained essentially unchanged at 117, 82, and 94 mmHg, respectively.

**Figure 6 cancers-17-00771-f006:**
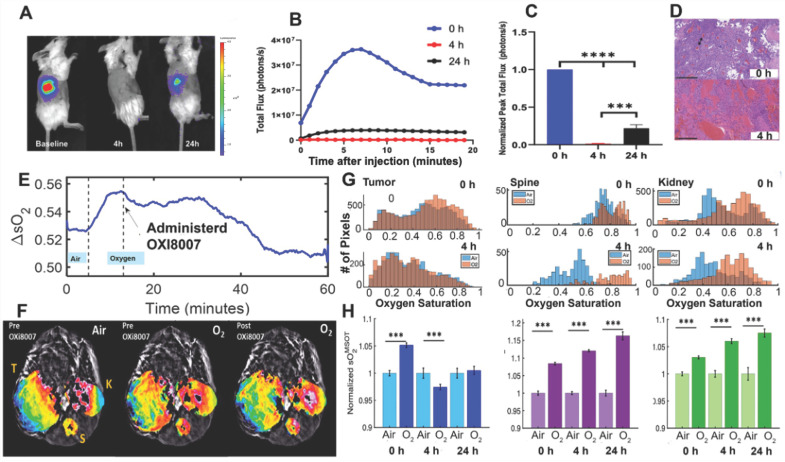
Acute effect of OXi8007 on orthotopic Renca-luc tumors in BALB/c mice. (**A**) BLI of Renca-luc tumor-bearing mouse at baseline (0 h), 4 h, and 24 h post-treatment of OXi8007 (250 mg/kg, IP). (**B**) Representative BLI dynamic curves at 0 h, 4 h, and 24 h. (**C**) Normalized group average (n = 8 mice) for peak total flux (photons/s) for 0 h, 4 h, and 24 h. *p* < 0.001 and 0.0001 indicated as *** and ****, respectively. (**D**) H&E-stained sections indicating congested tumor vasculature at baseline (black arrow) with extensive hemorrhage and vascular disruption 4 h post-treatment (bar 250 µm). (**E**) Mean tumor sO_2_ while breathing air (5 min), followed by O_2_ and in situ administration of OXi8007 (250 mg/kg, IP) using a catheter. (**F**) Transaxial cross-section photoacoustic image (800 nm) was superimposed with sO_2_ saturation maps of the tumor (T), kidney (K), and spine (S), while breathing air was followed by O_2_ and the administration of OXi8007 in situ while continuing to breathe O_2_ for 1 h. (**G**) Histograms of vascular sO_2_ in tumor, kidney, and spine while breathing air (blue) and oxygen (red) before and 4 h after the administration of OXi8007 (250 mg/kg, IP). (**H**) Mean oxygenation while breathing air and O_2_ at 0 h, 4 h, and 24 h post-treatment for tumor, spine, and kidney (n = 6). *** *p* < 0.001.

**Figure 7 cancers-17-00771-f007:**
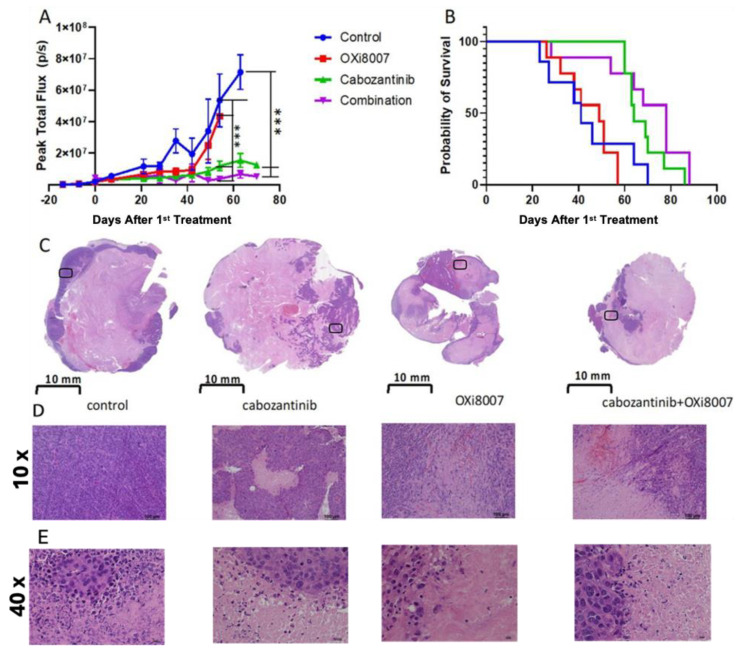
Response of orthotopic Renca−-luc tumors to treatment with OXi8007 and/or cabozantinib. (**A**) Tumor growth assessed by BLI (*** *p* < 0.001). (**B**) Kaplan−–Meier survival curve. Treatment groups were Group 1 (control) (n = 7, blue line), Group 2 (OXi8007) (250 mg/kg IP twice weekly; n = 9, red line), Group 3 (cabozantinib) (3 mg/kg oral daily; n = 9, green line), and Group 4 (combination) (n = 9, purple line). (**C**) Representative whole mount H&E-stained sections for each group showing extensive necrosis that was more prominent in treated mice and focal hemorrhage. (**D**) 10× magnified sections from sections in (**C**,**E**) 40× magnified sections from the 10× showing necrosis.

**Figure 8 cancers-17-00771-f008:**
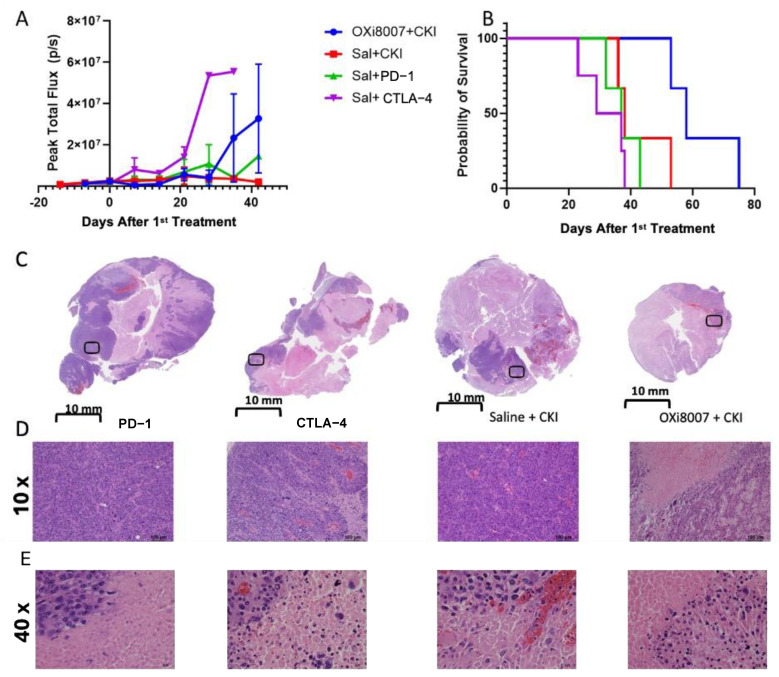
Renca-luc tumors treated with OXi8007 and checkpoint inhibitors (CKIs). (**A**) Growth curves for the individual groups. (**B**) Kaplan-Meier survival curves. Group 5 (combination) (OXi8007 + CKI (anti-CTLA-4 + anti-PD-1); n = 3, blue line), Group 6 (saline + CKI) (anti-PD-1 + anti-CTLA-4; n = 3, red line), Group 7 (saline + anti-PD-1) (n = 3, green line), and Group 8 (anti-CTLA-4) (n = 4, purple line). (**C**) Representative H&E-stained section for each treatment group showing extensive necrosis that was more prominent in mice treated with regiments containing CTLA-4 and/or OXi8007. (**D**) Magnified sections from boxes in (**C**,**E**) 40× magnified sections from the 10× showing necrosis.

**Figure 9 cancers-17-00771-f009:**
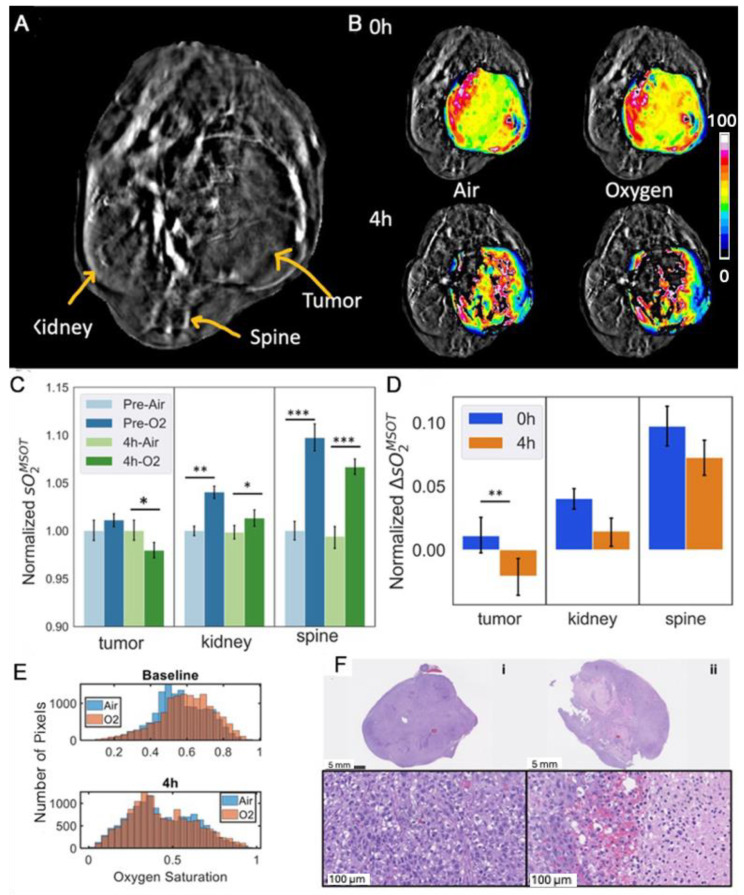
Patient-derived XP258 orthotopic kidney tumor model. (**A**) Transaxial MSOT (800 nm) cross-section of an NOD/SCID mouse showing tumor, spine, and kidney. (**B**) sO_2_ maps of tumor superimposed on 800 nm background map before and 4 h post-administering OXi8007 (IP) with respect to oxygen gas breathing challenge. (**C**) Oxygenation in tumor, kidney, and spine during O_2_ gas breathing challenge at 0 and 4 h post--administration of 250 mg/kg OXi8007 (n = 3). (**D**) Change in vascular oxygenation in tumor, kidney, and spine in response to oxygen gas breathing challenge before (blue) and 4 h after OXi8007 IP. (*** *p* < 0.001, ** *p* < 0.01 and * *p* < 0.05) (**E**) Histograms showing oxygenation distribution in one tumor upon administering the VDA and (**F**) H&E for those untreated (**i**) and treated (**ii**) with VDA after 4 h, showing hemorrhagic areas.

**Figure 10 cancers-17-00771-f010:**
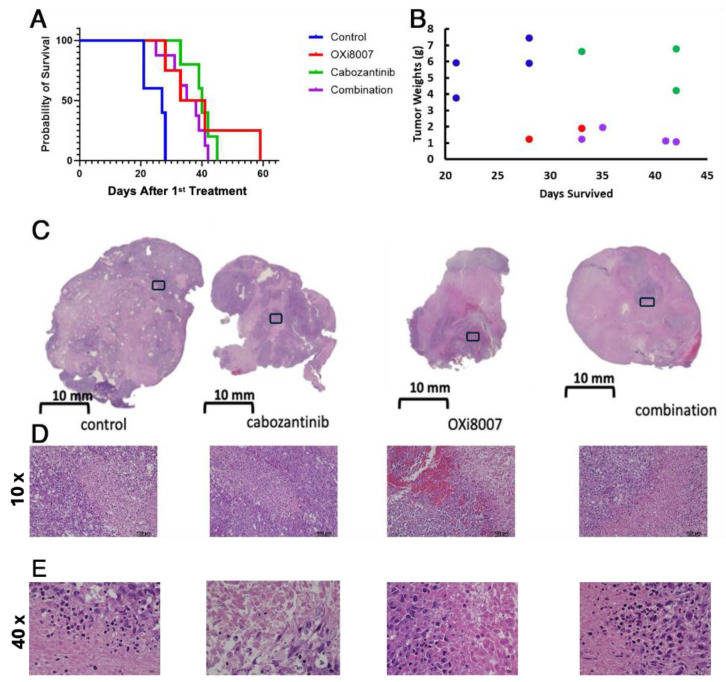
XP258 human tumor xenograft response to therapy. (**A**) Kaplan–Meier curves for control tumors (n = 5) and cohorts receiving OXi8007 (n = 4), cabozantinib (n = 5), and combination (n = 8). (**B**) Tumor weights compared with survival times. (**C**) H&E of representative XP258 tumors from each group at termination with more pronounced necrosis and hemorrhage observed in mice treated with OXi8007. (**D**) Magnified sections from boxes in (**C**,**E**) 40× magnified sections from the 10× showing necrosis.

**Table 1 cancers-17-00771-t001:** Cell growth inhibition.

Tumor Cells	Compound	IC_50_ (µM)
Renca	CA4	0.048 ± 0.002
	OXi8006	1.575 ± 0.153
	OXi8007	1.450 ± 0.001
Renca-luc	CA4	0.109 ± 0.023
	OXi8006	1.142 ± 0.097
	OXi8007	1.562 ± 0.048

**Table 2 cancers-17-00771-t002:** Pharmacokinetics of OXi8007 after OXi8007 (250 mg/kg) treatment.

OXi8007	T_1/2_ (h)	T_max_ (min)	C_max_ (ng/mg)	AUC (h × ng/ mg)
plasma	0.82	20	453 ^#^	511 ^#^
tumor and right kidney	2	60	14.99	36.31
left kidney	1.18	20	1.4	2.05
liver	1.1	20	13.4	12.67
brain	0.51	60	0.07	0.11
lungs	0.7	20	1.61	2.3
heart	0.61	20	1.62	1.32
spleen	0.65	60	0.99	1.41

^#^ Plasma C_max_ has units ng/µL, and AUC is h × ng/µL. T_½_ = half-life; T_max_ = time of the first occurrence of C_max_; C_max_ = maximum observed concentration; AUC = area under the concentration time curve from zero to time of last quantifiable concentration.

**Table 3 cancers-17-00771-t003:** Pharmacokinetics of OXi8006 after OXi8007 (250 mg/kg) treatment.

OXi8006	T_1/2_ (h)	T_max_ (min)	C_max_ (ng/mg)	AUC (h × ng/mg)
plasma	1.99	20	29 ^#^	47 ^#^
tumor and right kidney	4.23	60	2.39	10.24
left kidney	1.73	20	5.86	8.72
liver	2.25	20	9.16	18.19
brain	1.71	20	0.22	0.54
lungs	2.56	20	4.18	9.45
heart	2.02	20	5.31	9.04
spleen	1.72	20	4.45	9.88

^#^ Plasma C_max_ has units ng/µL, and AUC is h × ng/µL.

**Table 4 cancers-17-00771-t004:** Biodistribution of OXi8006-glucuronide after OXi8007 (250 mg/kg) treatment.

OXi8006 Glucuronide	T_1/2_ (h)	T_max_ (h)	C_max_ (counts × 10^−6^) ^+^	AUC (h × counts × 10^−6^) ^+^
plasma	1.95	1	1266	3490
tumor	5.87	4	11.65	118.22
left kidney	3.75	1	87.2	180.6
liver	3.37	1	28.23	128.7
brain	1.15	1	0.95	4.13
lungs	2.05	1	20.87	69.33
heart	1.61	1	9.8	24.52
spleen	2.2	1	6.33	18.39

^+^ In the absence of precise calibration for the glucuronides, data for C_max_ are presented as counts per tissue sample (each 40 mg) and counts per mL for plasma.

## Data Availability

Data can be shared on request.

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
