# Peer review of "Evaluating Therapeutic Efficacy of the Vascular Disrupting Agent OXi8007 Against Kidney Cancer in Mice [Author-notes fn2-cancers-17-00771]"

_cancers, 2025, doi:10.3390/cancers17050771_

Round 1
Reviewer 1 Report
Comments and Suggestions for Authors
The authors investigate the treatment effects of OXi8007, a vascular disrupting agent (VDA), combined with cabozantinib, anti-PD-1, and anti-CTLA-4 agents in kidney cancer.
Some comments are listed below.
1. Supplemental data is missing.
2. For the in vivo study, how the mice (tumor xenograft) were randomized to each group needs to be described in the methods. In addition, the authors described “orthotopic XP258 human tumor xenografts, where treatment started 9-14 days post tumor implantation”. How was the time frame decided, and how were these mice randomized into different groups?
3. Need the detailed route of how OXi8007, anti-PD-1 and anti-CTLA-4 solutions were administered to the mice.
4. Provide data comparing the difference in mouse body weight in different treated groups to demonstrate the potential toxicity.
5. The wound healing assay demonstrated that OXi8007 reduced the migration of Renca cells. Considering that the OXi8007 is a VDA, what are the potential mechanisms of OXi8007 to reduce cancer cell migration (provided in the discussion section)?
6. What is the rationale for monitoring the mouse blood pressure in this study?
7. Figure 6: Some fonts are too small to visualize.
8. Figures 7D, 8D, 9F, and 10D: The magnification of images is too low to visualize the necrosis lesion. The authors can consider using the necrosis marker or staining to demonstrate the treatment's effect (quantification) in different groups.
9. Figures 9F and 10D: The authors can consider performing IHC with human-specific Ku70 antibody to demonstrate the tumors were patient-derived xenograft (PDX).
10. Does any renal tumor metastasis to other mouse organs (lung, liver…)?
11. Lines 485-487: “In some cases, OXi8007 yielded a tumor growth delay as a monotherapy, and in other cases enhanced combination treatment. It is reassuring that OXi8007 caused no overt toxicity when dosed at 250 mg/kg over many weeks”. Need to describe the precise case conditions, and the time frame.
12. Provide a figure to summarize this study.
13. Line 55: mm3
14. Line 77: mTOR
Reviewer 2 Report
Comments and Suggestions for Authors
Dear Authors,
I reviewed your paper with interest.
This article provides a detailed examination of the therapeutic potential of OXi8007, a vascular disrupting agent, in combating kidney cancer through preclinical studies in mice. The comprehensive methodology and robust data presentation underscore the study’s scientific rigor.
Despite its strengths, the manuscript requires some improvements before publication. Therefore, I have compiled a list of comments for your consideration.
1. While the article targets a specialized audience, simplifying technical jargon in sections like the introduction could broaden its accessibility. Provide a clearer explanation of complex processes, such as the mechanism of vascular disruption by OXi8007.
2. The discussion largely reiterates the results. Expanding on the clinical implications, limitations, and challenges in translating these findings to human studies would enrich the article.
3. The narrative should better integrate figures and tables, explicitly referencing them to aid comprehension.
4. Although OXi8007 is compared to cabozantinib and CKIs, a deeper analysis of how it stands relative to other VDAs in preclinical or clinical trials would provide better context.
5. Some statistical outcomes are presented without adequate explanation of their significance. Brief descriptions of the statistical methods and interpretations would be beneficial.
6. Addressing the accessibility of such VDAs in resource-constrained settings would add a valuable dimension to the study’s impact.
7. Minor grammatical and typographical errors were noted throughout the text. A thorough editorial review is recommended to ensure precision and professionalism.
8. Potential Additions:
- Future Directions: a dedicated section on potential dosing schedules, nanoparticle delivery systems, or alternative applications of OXi8007 could stimulate further research.
- Comparative Toxicology: while the study reassures OXi8007’s tolerability, more detailed toxicity comparisons with other VDAs could strengthen its safety profile.
The article is a significant contribution to the field, demonstrating the potential of OXi8007 in the treatment of kidney cancer. Incorporating the recommendations above would enhance the paper’s clarity, depth, and practical relevance, making it a more comprehensive resource for both researchers and clinicians.
Comments on the Quality of English LanguageMinor grammatical and typographical errors were noted throughout the text. A thorough editorial review is recommended to ensure precision and professionalism.
Round 2
Reviewer 1 Report
Comments and Suggestions for Authors
- Line 49: There are some words and sub-figures too small to visualize in the Summary Figure.
Author Response
1. Line 49: There are some words and sub-figures too small to visualize in the Summary Figure.
We fixed the font sizes in the figure and enlarged the small images. Please see the attached figure.
2. Please note that we changed the citation #68.
